

# A bigura-based real time sentiment analysis of new media

Haili Xu

Guangzhou Huashang College, Guangzhou, Guangdong, China

## ABSTRACT

Public opinion mining is an active research domain, especially the penetration of the internet and the adoption of smartphones lead to the enormous generation of data in new media. Thus generation of large amounts of data leads to the limitation of traditional machine learning techniques. Therefore, the obvious adoption of deep learning for the said data. A multilayer BiGura modal-based technique for real-time sentiment detection is proposed. The proposed system is analysed on different viral incidents such as Gaza's invision. The exact case scenario is as follows "Taking Israel's demand for millions of people from northern Gaza to migrate to the south". In the experiment, the highest accuracy of the model in evaluating text content emotions and video content emotions reached 92.7% and 86.9%, respectively. Compared to Bayesian and K-nearest neighbour (KNN) classifiers, deep learning exhibits significant advantages in new media sentiment analysis. The classification accuracy has been improved by 3.88% and 4.33%, respectively. This research identified the fidelity of real-time emotion monitoring effectively capturing and understanding users' emotional tendencies. It can also monitor changes in public opinion in real-time. This study provides new technical means for sentiment analysis and public opinion monitoring in new media. It helps to achieve more accurate and real-time monitoring of public opinion, which has important practical significance for social stability and public safety.

## INTRODUCTION

In the digital media environment, the emergence of new media provides convenient ways for the public to obtain and express information, greatly promoting the diversification and personalization of information dissemination (*Hafzullah Tuncer, 2021*; *Jin et al., 2021*). However, what comes with it is how to accurately understand and analyze the emotional and public opinion tendencies brought about by these massive amounts of information. This has significant implications for policymakers, marketers, and even the general public. An effective solution is to use deep learning technology for sentiment analysis and public opinion monitoring (*Su, 2021*; *Middya, Nag & Roy, 2022*). Deep learning technology, with powerful nonlinear mapping and pattern recognition capabilities, has become a powerful tool for processing complex, high-dimensional, and unstructured data, especially demonstrating excellent performance in data analysis fields such as text, image, and speech

Corresponding author
Haili Xu, xuhaili2023@163.com

(*Hafzullah Tuncer, 2021*; *Hui, 2021*). Therefore, deep learning technology for emotional analysis and public opinion monitoring of new media content has high research and practical value. The research aims to explore emotional analysis and public opinion monitoring methods in the new media environment through deep learning technology. Firstly, deep learning technology is used for feature extraction and sentiment classification of new media content. Based on this, public opinion tendency is predicted.

Deep learning is continuously enhancing the abilities of automated systems for diverse data needs including sentiment detection, helps to timely identify potential risks, guide public opinion, and respond to crises. The contribution lies in providing a new and effective method for sentiment analysis and public opinion monitoring in new media.

Moreover, this method has important application value in understanding the public's emotional attitudes towards specific topics and predicting and controlling public opinion risks. In addition, this also provides new research perspectives and methodological references for the relevant theories of new media sentiment analysis and public opinion monitoring (*Su, 2021*). The research has four parts. The first part is an overview of new media sentiment analysis and public opinion monitoring technology based on deep learning. The second part is the research on new media sentiment analysis and public opinion monitoring technology based on deep learning. The third part is the experimental verification for the second part. The fourth part is a summary and points out the shortcomings.

## RELATED WORKS

With the rapid development of new media, the application of deep learning in the emotional analysis of new media has become increasingly widespread, attracting wide attention from the academic community. *Chen & Zhang (2023)* applied edge computing and deep learning models to the emotion recognition model for non-profit organizations (*Middya, Nag & Roy, 2022*; *Hafzullah Tuncer, 2021*; *Hui, 2021*; *Chen & Zhang, 2023*). The purpose is to understand the evolutionary mechanism of online public opinion in sudden public events. The research results reveal that non-profit organization text annotation based on emotional rules can achieve good recognition performance. The improved convolutional neural network has significantly better recognition performance than traditional support vector machines. This work provides a technical basis for non-profit organizations to scientifically handle sudden public events (*Chen & Zhang, 2023*). *Manohar & Logashanmugam (2022)* proposed a new method for speech emotion recognition. The deep learning model is used to process preprocessed public speech emotion recognition datasets. When the learning rate is 85, the classification accuracy of the model is 3.15%, 5.37%, 4.25%, and 4.81%, higher than the particle swarm optimization algorithm, Grey Wolf Optimizer (GWO) algorithm, Whale Optimization Algorithm (WOA) algorithm, and the Dynamic Hybrid Optimization Algorithm (DHOA), respectively. It proves the superior performance of the model in speech emotion recognition. *Zhang, Dai & Zhong (2022)* proposed a deep learning computing method and an emotion recognition method. A public sentiment network communication model is established. The proposed Recurrent Neural Network-Convolutional Neural Network (RNN-CNN) structure can reduce the waiting time by

about 20% compared to traditional models. The algorithm accuracy is improved by at least 3.1%. It can accurately reflect the emotional state of the public, providing a practical basis for the application of AI technology in online public opinion judgment. *Schoneveld, Othmani & Abdelkawy (2021)* proposed a deep learning-based audiovisual emotion recognition method to achieve an understanding of complex human behaviour. This method utilizes a model-level fusion strategy to fuse deep feature representations of audio and visual modalities. Recurrent neural network is used to capture temporal dynamics. The research results show that this method outperforms existing technologies in predicting the potency of the RECOLA dataset. It performs well on the AffectNet and Google facial expression comparison datasets. *Wang, Luo & Song (2021)* proposed a hybrid neural network model based on Recurrent Neural Network-Convolutional Neural Network (RNN-CNN) to solve the precise classification. The accuracy is 92.8%, the minimum loss rate is 0.2, and the trend is stable. The model can obtain more semantic information between texts. It can also better capture the dependency relationships.

Although many studies have explored the application of deep learning in public opinion monitoring, there are still many unresolved issues and areas for improvement in this field. *El Barachi et al. (2021)* proposed a novel framework. A complex bidirectional long short term memory (LSTM) classifier is used for real-time evaluation of the viewpoints of well-known public figures and their followers. The results show that the classifier has an accuracy of over 87% in identifying multiple emotions and viewpoints. The recognition accuracy of negative emotions is higher. *Zheng & Xu (2021)* proposed a deep learning-based facial detection and tracking framework. This framework integrates the SENResNet model based on a squeezing excitation network and residual neural network, as well as a face-tracking model based on a regression network. The SENResNet can accurately detect facial information and provide an initialization window for facial tracking. Numerous experimental results have shown that this framework outperforms existing technologies in accuracy and performance. *Wang & Gao (2023)* proposed a solution based on deep learning (*El Barachi et al., 2021*). A system framework including text extraction, keyword extraction, and sentiment analysis modules is designed. An information extraction model is constructed using convolutional neural networks. By calculating the global Mutual Information (MI) values of text items and categories and inputting them into the model, information extraction results are obtained. The system has high extraction accuracy and fast extraction time. *Keivanlou-Shahrestanaki, Kahani & Zarrinkalam (2022)* proposed a deep-learning neural network architecture. This architecture explores the adaptive effects of different attention mechanisms, generating non-satirical posts with the same meaning as the original satirical posts. Numerous experimental results have demonstrated the effectiveness of this method in explaining satirical articles, especially when dealing with long posts.

In summary, deep learning has made significant contributions to new media sentiment analysis and public opinion monitoring, particularly demonstrating outstanding capabilities in processing large-scale, complex, and unstructured data. However, existing methods face challenges in dealing with complex, multi-semantic, and emotional new media content. Future research needs to develop more complex models to optimize the

understanding and capture of emotional expression. In addition, empirical research is needed to verify the effectiveness and practicality of deep learning in the application of new media sentiment analysis and public opinion monitoring. Despite the challenges, the application prospects of deep learning-based new media sentiment analysis and public opinion monitoring remain broad (*Zheng & Xu, 2021*). There is extensive application space in public opinion risk warning, public opinion analysis, and marketing decision-making.

## RESEARCH METHOD

New media sentiment analysis and public opinion monitoring are key links in data mining. Deep learning technology has shown outstanding performance in processing large-scale data and extracting emotional information (*Zhang, Dai & Zhong, 2022*). The attention convolutional neural network conditional random field word segmentation model combines attention mechanism and convolutional neural network to improve word segmentation accuracy and efficiency. The emotion analysis model based on aspect information utilizes aspect information to deepen the understanding and analysis of the emotional polarity of text (*De Martino & Netti, 2020*). The combination of the two models can further improve the accuracy and depth of new media sentiment analysis and public opinion monitoring, providing accurate references for decision-making.

### Construction of the attention convolutional neural network conditional random field word segmentation model

The attention convolutional neural network conditional random field word segmentation model is a deep learning model that combines the attention mechanism and convolutional neural network to improve accuracy and efficiency. The attention mechanism can focus on key vocabulary with important information. Convolutional neural networks can effectively capture the contextual relationships between words. Through this combination, the model can more accurately segment new media texts, providing more accurate input information for subsequent sentiment analysis and public opinion monitoring (*Li & Xu, 2020*; *Seokhoon, Jihea & Young-Sup, 2023*). The overall framework diagram of the attention convolutional neural network conditional (ACNNC) model is displayed in Fig. 1.

In Fig. 1, the ACNNC model is composed of five levels. The order is an embedding layer, attention layer, CNN layer, fusion layer, and CRF layer. Firstly, the embedding layer generates 128-dimensional word vectors through training, which will serve as inputs for subsequent levels. Then, the word vector is input to the attention layer and CNN layer. The former is responsible for learning the overall features of the sequence. The latter learns the positional and local features of the sequence. Next, the fusion layer integrates the overall, local, and positional features obtained from the above learning. Finally, the integrated features are decoded at the CRF layer to complete the construction of the model (*Hou, Wang & Wang, 2023*).

The embedding layer converts the *i*th word in a sentence into a word vector. Then, it is trained to obtain the word vector matrix. The size of the word vector matrix is determined by the effective length of the vocabulary in the corpus dataset and the dimension of the input word vector. Each word can be transformed into a corresponding word vector

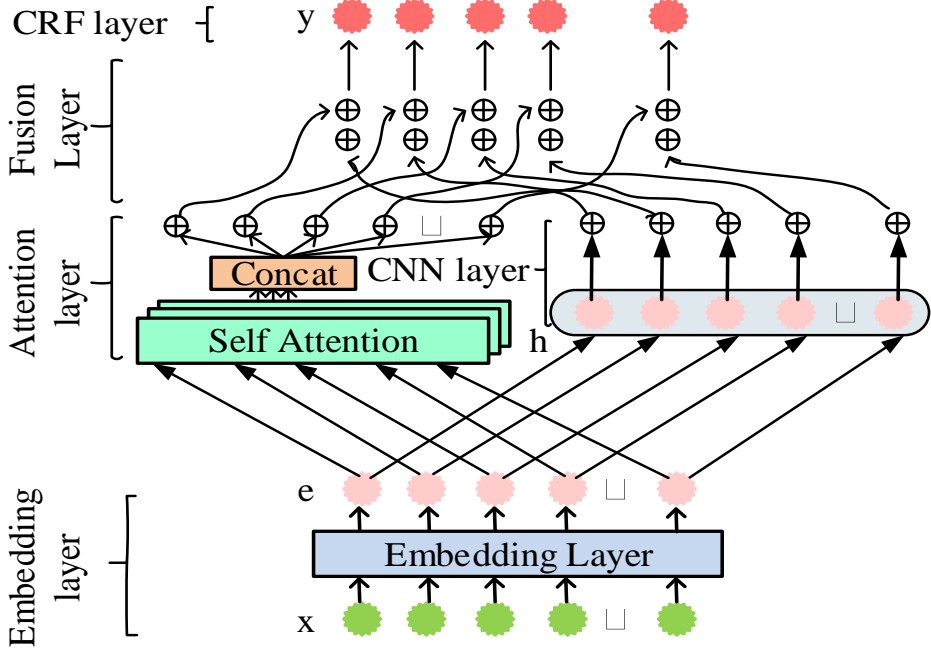

**Figure 1** Overall framework diagram of the ACNNC model.

representation through a word vector matrix. The embedding layer is displayed in Eq. (1).

$$e_i = X \times v_i \tag{1}$$

In Eq. (1), $X$ is the word vector matrix. $|N|$ is the effective length of the vocabulary in the corpus dataset. $v_i$ is a One-hot vector of size $|N|$. The sentence $x$ isconverted to $S$, as shown in Eq. (2).

$$S = (e_1, e_2, \ldots, e_n) \in R^{|N| \times d}. \tag{2}$$

In Eq. (2), $d$ stands for the dimension of the input word vector. The individual attention is shown in Eq. (3).

$$head_i = Attention\left(QW_i^Q, KW_i^K, VW_i^V\right). \tag{3}$$

In Eq. (3), $Q, K, V$ stand for the same value. The attention layer adopts a multi-head self-attention mechanism to reduce the possible random errors caused by the single-head attention mechanism and further improve the accuracy. Unlike conventional attention mechanisms, the dependency of self-attention mechanisms lies in themselves. That is, the query, key, and value are all the same value. The multi-head self-attention mechanism is shown in Fig. 2.

In Fig. 2, the query, key, and value are first linearly mapped, followed by eight individual attention calculations. Finally, the results are concatenated. The characteristic of this self-attention mechanism is that the queries, keys, and values are all the same value. In this way,

**Peer**J Computer Science

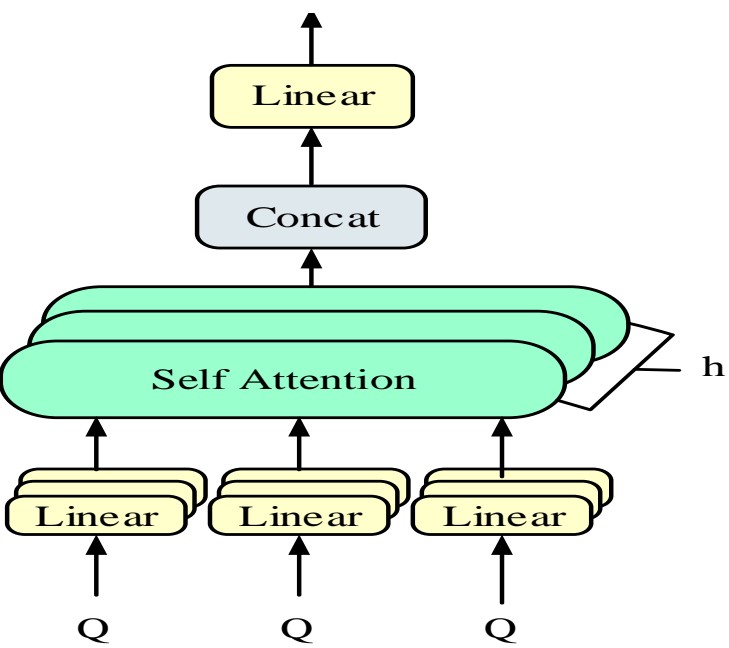

**Figure 2** Multi-head self-attention mechanism.

the multi-head self-attention mechanism can improve model accuracy. In the CNN layer of the model, only the convolutional layer and output layer are included. Input sentences processed by the embedding layer are fed into the convolutional layer for convolutional operations (*Fang et al., 2022*; *Yang & Song, 2022*). The convolution operation is shown in Eq. (4).

$$x_i^k = \sigma\left(\sum (X[i:i+k] \circ H_k) + b\right).$$ (4)

In Eq. (4), $\sigma$ stands for the activation function. $\circ$ is a point multiplication operation. $X[i:i+k]$ is a sequence of word vectors. $H_k$ is a convolutional kernel of size $k$. The fusion layer integrates the features of the attention layer and the CNN layer, generating feature parameters containing overall, local, and positional information and then inputting them into the CRF layer. This model utilizes a vector concatenation strategy for feature fusion. CRF is a discriminative undirected graph model that can consider changes in data content and labels. It is widely used in tasks such as Chinese word segmentation, named entity recognition, and part of speech tagging. The processed sentences in the model are output through the fusion layer and then input into the CRF layer to obtain the results of sequence annotation. After indexing and standardizing the scores of sequence annotations, the final probability value is obtained. The calculation of $S_t$ is shown in Eq. (5).

$$S(y|x) = \sum_{i,k_1} \theta_{k_1} t_{k_1}(y_{i-1}, y_i, x, i) + \sum_{i,k_2} \mu_{k_2} s_{k_2}(y_i, x, i).$$ (5)

In Eq. (5), $O_t$ represents the character after passing through the fusion layer. $t_{k_1}(y_{i-1}, y_i, x, i)$ is the transfer feature function. $s_{k_2}(y_i, x, i)$ is the state feature function.

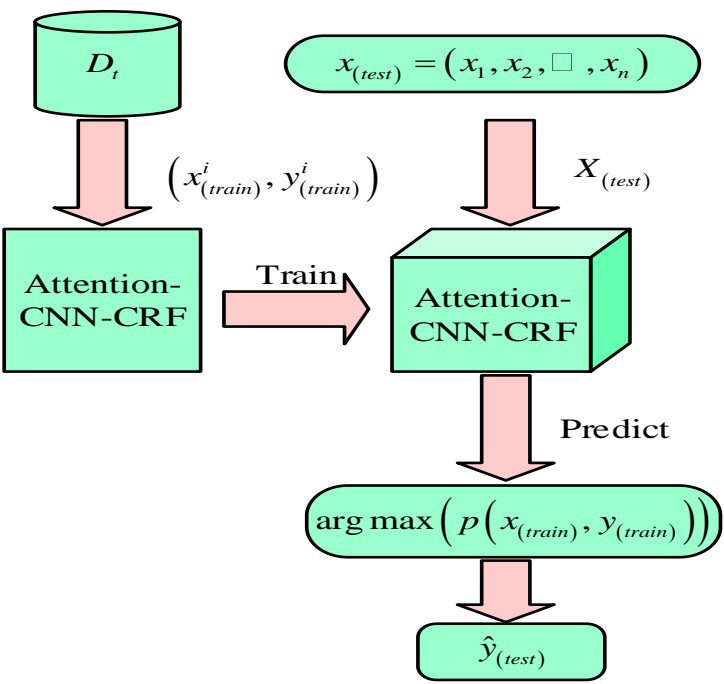

**Figure 3** Process diagram of word segmentation sequence.

$\theta_{k_1}$ and $\mu_{k_1}$ are the model parameters to be estimated, respectively. The probability value is shown in Eq. (6).

$$P(\hat{y}|x) = \frac{1}{Z(x)}\left(\theta_{k_1} t_{k_1}(y_{i-1}, y_i, x, i) + \sum_{i, k_2}\mu_{k_2} s_{k_2}(y_i, x, i)\right). \tag{6}$$

In Eq. (6), $P(\hat{y}|x)$ is the probability value. The normalization factor is the sum of all possible marker sequences, as shown in Eq. (7).

$$Z(x) = \sum_{y}\exp\left(\sum_{i, k_1}\theta_{k_1} t_{k_1} + \sum_{i, k_2}\mu_{k_2} s_{k_2}(y_i, x, i)\right). \tag{7}$$

Equation (7) $Z$ is the normalization factor. The task of word segmentation sequence annotation is to select the most likely marking sequence for each character in a given sentence. The markers include the Begin (B), Middle (M), End (E), and Single (S). The sequence annotation model is used for training and prediction, calculating conditional probabilities. Then, the sequence with the highest probability is output to obtain the segmentation result. The process diagram of the word segmentation sequence is shown in Fig. 3.

In Fig. 3, the word segmentation process includes four stages: preprocessing, model training, prediction, and post-processing. During the preprocessing stage, special symbols, punctuation, and stop words are removed to reduce model complexity. During the model
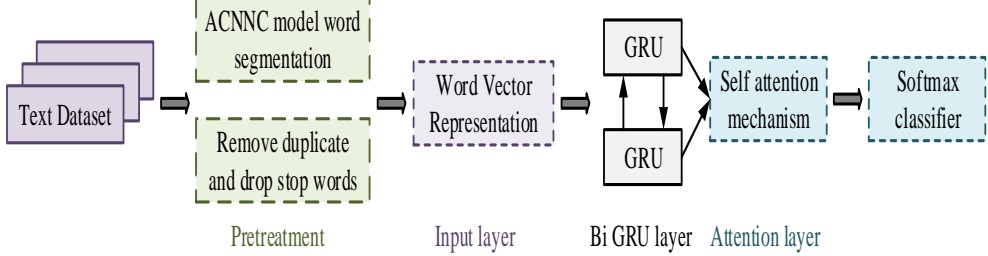

**Figure 4** **BiGRUA model flowchart.**

training phase, a large amount of annotation data is used to train sequence annotation models, such as HMM, BiLSTM, *etc.*, to generate the most likely label sequence for each character or sub-word. In the prediction stage, a trained model is used to segment new sentences and output a tag sequence representing the most likely markers for each character or sub-word. The post-processing stage converts the tag sequence into the actual segmentation result. The specific methods for each stage depend on task requirements and resource constraints.

## Construction of an emotional analysis model based on aspect information

On the basis of the ACNNC model, this study further explores the construction of an emotion analysis model based on aspect information. Emotional analysis faces complex contextual and subjective challenges in extracting emotional tendencies from texts. To obtain emotional information more accurately, aspect information is introduced for more in-depth text fine-grained emotional analysis. The BiGRUA model flowchart is shown in Fig. 4.

In Fig. 4, the construction of the model includes five stages, preprocessing, input layer, Bidirectional Gated Recurrent Unit (BiGRU) layer, attention layer, and classification layer. Firstly, in the preprocessing stage, the original text data is cleaned and standardized to remove irrelevant information and extract effective features. Next, the input layer trains the data through the Word2vec model to obtain a word vector matrix. It can transform the original text data into a sequence of word vectors with certain semantic information. Next, the training set and corresponding training labels are fed into the BiGRU layer. At this stage, the model learns contextual information in the text through forward and backward GRU and updates model parameters to better capture semantic information in the text. The output of the BiGRU layer is fed into the attention layer. The attention mechanism focuses on the most important parts of the text, thereby extracting more representative features. Finally, the task of the classification layer is to classify emotions based on the output of the attention layer. Through this process, the model can extract useful features

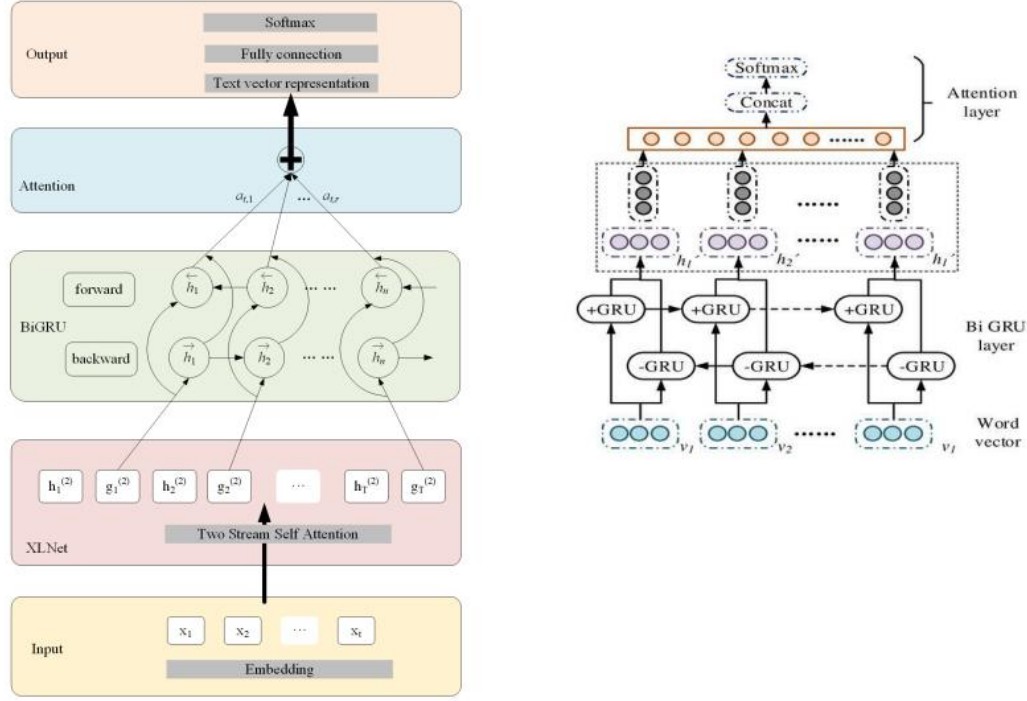

**Figure 5   BiGRUA model framework diagram (*Han et al., 2023*).**

from the original text data and perform accurate emotional analysis. The BiGRUA model framework diagram is shown in Fig. 5 and its clever setting is compared with other models.

In Fig. 5, BiGRUA is proposed for sentiment analysis of text containing multiple words, which integrates aspect information. It uses GRU instead of LSTM for feature learning. The attention mechanism is used to allocate aspect information weights. Finally, sentiment analysis is achieved through a softmax classifier. The BiGRU layer solves the gradient vanishing and long-distance semantic capture in recurrent neural networks by using GRU units in update and reset gates. It combines positive and negative implicit states, which can fully learn reverse semantics and obtain more complete contextual information. The BiGRU layer structure is shown in Fig. 6.

In Fig. 6, the unit structure of GRU plays a key role in the sentiment analysis model based on aspect information. Based on reset and update gates, the model can handle long-distance semantic dependencies and adapt to complex sentiment analysis tasks. The reset gate determines the old information that the model should retain or discard before further processing new information. The update gate is responsible for how to integrate new and old information to generate the latest status. This mechanism enables GRU to effectively alleviate the gradient disappearance while capturing long-term dependencies. In the emotion analysis model based on aspect information, the unit structure of GRU has crucial impacts on the accuracy of the model due to the special design and critical role. Based on the clever setting of reset and update gates, this model can handle the complex task of finding semantic associations in long-distance contexts, thus adapting to the

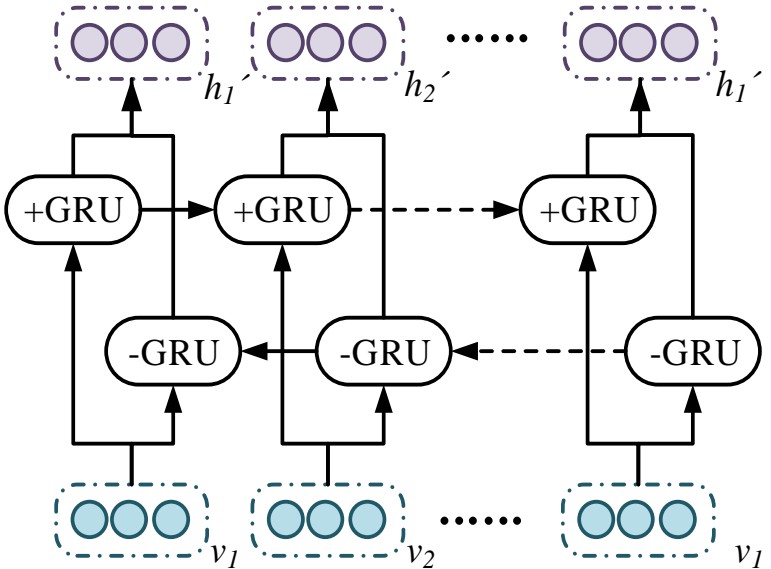

**Figure 6   BiGRU layer structure.**

complexity of sentiment analysis. This clever setting may be described from its connectivity with self-attention and Bayes condition probabilities. For a clear understanding, an the architecture of BiGRU is used for text emotion detection and visual emotion detection is given in the Fig. 5. The purpose of resetting the gate is to determine which old internal states should be retained or discarded before further processing new information. This is a key mechanism. GRU can flexibly adjust the internal state based on new input and context, thereby effectively capturing and understanding long-distance dependencies in text. The update gate is responsible for fusing new and old information to generate the latest status. This mechanism is implemented through a carefully designed gating mechanism. It can fuse and update new and old information appropriately based on the characteristics and context of the input text. This design enables GRU to effectively alleviate the gradient vanishing while capturing long-term dependencies, thereby improving the stability and accuracy of the model.

The output of the BiGRU layer at a time $t$ is displayed in Eq. (8).

$$h_t' = \left[ \overrightarrow{h}_t, \overleftarrow{h}_t \right]. \tag{8}$$

In Eq. (8), $h_t'$ is the output layer. The attention layer is to distinguish the importance of different parts in a sentence, as the influence of target words and viewpoint words on the emotional orientation in the text is different. The attention mechanism is applied to assign different weights to different parts, highlighting the contribution of different aspect words to emotional tendencies. The input is the output of the BiGRU layer, which is transformed into a new hidden vector through a multi-layer perceptron. Then it is calculated with the context vector to obtain the weight value. This context vector is a high-dimensional vector

used to determine the importance of words in a sentence. The attention layer is displayed in Eq. (9).

$$u'_t = \tanh\left(W_w h'_t + b_w\right)$$
$$\alpha_t = \frac{\exp\left(u'^T_t u'_v\right)}{\sum_t \exp\left(u'^T_t u'_v\right)} . \tag{9}$$

Equation (9), $u'_t$ is the new hidden vector. $\alpha_t$ is the weight value. $u'_v$ is a high-dimensional vector used to determine the importance of words in a sentence. The average target word vector is shown in Eq. (10).

$$V_{at} = \frac{1}{m}\sum_{i=1}^{m} e_i. \tag{10}$$

Equation (10) $m$ is the target word in the sentence. $V_{at}G$ is the target word vector. The fusion vector is shown in Eq. (11).

$$P_i = h'_t + V_{at}. \tag{11}$$

In Eq. (11), $h'_t$ is the hidden vector of the output. $P_i$ is the fusion vector. The emotional features of attention weights that integrate aspect information are shown in Eq. (12).

$$C' = \sum_t \alpha_t h'_t. \tag{12}$$

In Eq. (12), $C'$ integrates the emotional features of aspect information attention weights. The classification layer inputs the features captured by the attention layer into the softmax classifier for classification. The activation function adopts the sigmod function, as shown in Eq. (13).

$$\widetilde{y} = soft\max\left(W_{C'} C' + b_{C'}\right). \tag{13}$$

In Eq. (13), $W_{C'}$ is the weight matrix. $b_{C'}$ is the offset value. $\widetilde{y}$ is the classification result. The cross entropy loss function is shown in Eq. (14).

$$L = -\frac{1}{N}\sum_{r=1}^{m}\sum_{q=1}^{C} y_{rq} \log\left(\widetilde{y}_{rq}\right) + \lambda \|\theta\|_2. \tag{14}$$

In Eq. (14), $y$ and $\widetilde{y}$ stands for the actual label values and predicted label values. $C$ is the number of label categories. $\lambda$ is the $L_2$ regularization coefficient.

## New media sentiment analysis and public opinion monitoring technology analysis based on deep learning

The rapid development and widespread application of new media have enabled the rapid acquisition and dissemination of large-scale public sentiment tendencies and public opinion dynamic information. Deep learning, as a complex and powerful machine learning technology, has been widely applied in sentiment analysis and public opinion monitoring. The application effect of deep learning in new media sentiment analysis is discussed in detail. The performance of new media public opinion monitoring technology is compared, providing theoretical reference and practical guidance for research in related fields.

**Table 1  Set parameters.**

| Parameters | Description | Value |
| --- | --- | --- |
| Word2Vec Dimension | Size of word vectors | 100 |
| Hidden Layer Size | Size of hidden layer | 100 |
| Attention Weight Length | Length of attention weights | Varies |
| Optimizer | Models optimizer | Adam |
| Learning Rate | Learning rate | 0.001 |
| Dropout | Dropout rate | 0.2 |
| Epochs | Number of iterations | 15 |
| Batch_size | Batch size | 64 |
| L2 Regularization Coefficient | L2 regularization coefficient | 0.0001 |

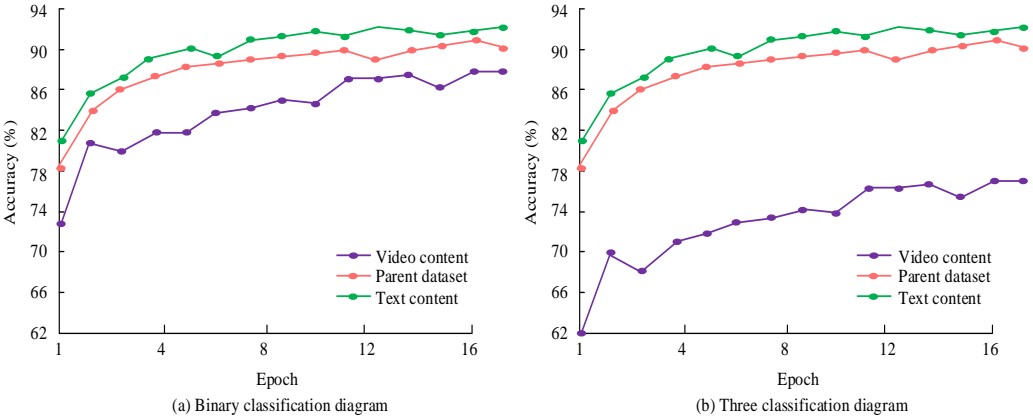

(a) Binary classification diagram

(b) Three classification diagram

**Figure 7  Experiment on two and three classifications of new media content.**

## The application effect of deep learning in new media sentiment analysis

In the new media sentiment analysis, the application effect of deep learning has attracted much attention. By constructing precise models and algorithms, deep learning can accurately identify and understand public emotions, providing valuable emotional feedback for enterprises and organizations. Table 1 displays the parameters.

To better demonstrate the practical application effect of deep learning in emotional analysis, a study collected new media content from a certain self-media platform as the parent dataset and divided the dataset into two sub-datasets: text content and video content. Based on these two sub-datasets, experiments were conducted on binary and ternary classification, as shown in Fig. 7.

During the experiment, the study conducted multiple iterations based on different parameter settings and recorded the accuracy of the model after each iteration. All parameter settings and corresponding accuracy are shown in Table 1. In Fig. 7, in the binary classification experiment, the model achieved the highest recognition accuracy of 92.7% in text-content emotional expression and 86.9% in video-content emotional expression

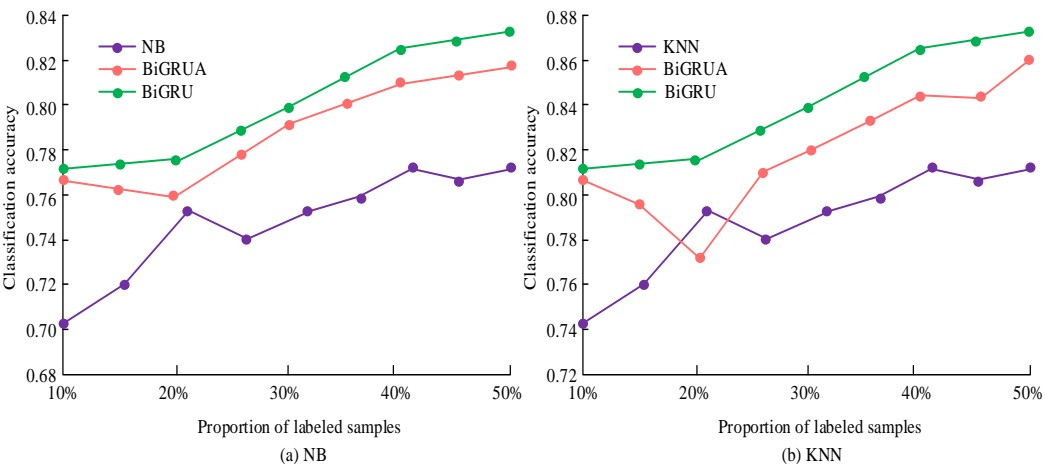

**Figure 8** **Training classification accuracy based on NB and KNN classifiers.**

evaluation. The accuracy of recognizing emotions in text content gradually increased from 82.9% in the first iteration to the highest in the 11th iteration and fluctuated after that. The accuracy of video content emotion recognition fluctuates and increases, reaching its highest after 15 iterations. When analyzing new media content datasets, the highest accuracy can reach 91.4%. In the three classification experiments, the results were similar to those of the two classifications. The highest accuracy of text content emotion expression recognition is 90.8%, and the highest accuracy of the parent dataset is 89.0%. The accuracy of emotional expression evaluation in video content is relatively low. The highest accuracy of the ternary classification is lower than that of the binary classification. The reason may be that the increase in classification types led to a decrease in accuracy. Deep learning achieved high accuracy in emotional analysis of text content and video content, whether it is for binary or ternary classification. These experimental results fully demonstrated the superior performance of deep learning in new media sentiment analysis. It further confirmed the effectiveness of the parameter settings in Table 1. The accuracy of deep learning in new media sentiment analysis is shown in Fig. 8.

In Fig. 8, deep learning demonstrates significant advantages in the analysis of new media emotions. The classification accuracy is improved by 3.88% and 4.33% compared to Bayesian and KNN classifiers, respectively. After optimization, the accuracy is further improved to 6.74% and 6.97%. However, the performance of Bayesian models deteriorates when labelled data accounts for 40%. KNN performs excellently when labelled data accounts for 20%. Deep learning has improved the accuracy of sentiment classification and assisted in real-time public opinion monitoring, which is of great value for policy formulation and marketing.

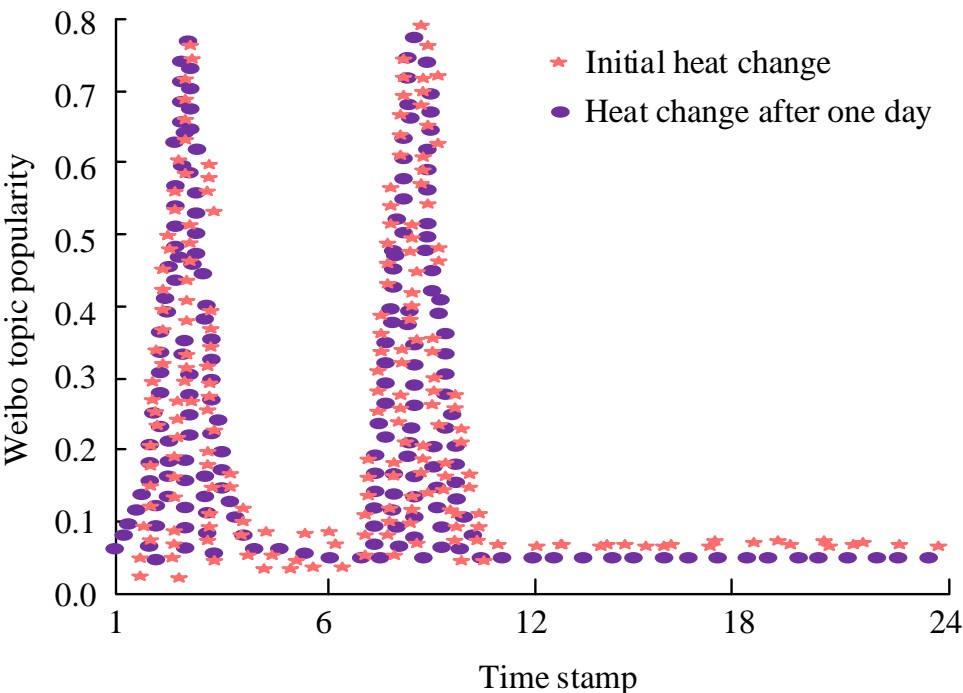

**Figure 9** Score chart of the popularity of the topic "Israel demands millions of people from northern Gaza to migrate to the south".

## Experimental results and analysis of deep learning in public opinion monitoring

The sentiments expressed *via* the Sina Weibo API center around the notion of Israel urging millions of individuals from northern Gaza to relocate southward. Following data preprocessing, the information is structured in JSON format. Using this scenario as a case in point, an examination is conducted on the comprehensive metrics encompassing topic popularity, sentiment ratio within comments, and the intensity of emotions expressed. In light of Israel's relocation request, there has been a notable escalation in the intensity of public discourse, underscoring the necessity for vigilant public opinion monitoring, particularly in gauging both the fervor surrounding an incident and the prevailing emotional inclinations among the populace. The intensity of discussion, as reflected in the heat score pertaining to the topic "Israel requires millions of people from northern Gaza to migrate to the south," is illustrated in Fig. 9.

Figure 9 depicts the emotional landscape surrounding the topic of "Israel requiring millions of people from northern Gaza to migrate to the south." Initially, following Israel's relocation proposal, there was a surge in public sentiment and the intensity of discussion. However, over time, the frequency of public discourse decreased, and emotions gradually subsided, returning to a state of calm. This underscores the significance of robust public opinion monitoring models, which necessitate keen observation of both the momentum of events and the prevailing sentiments among the public. The emotional orientation map

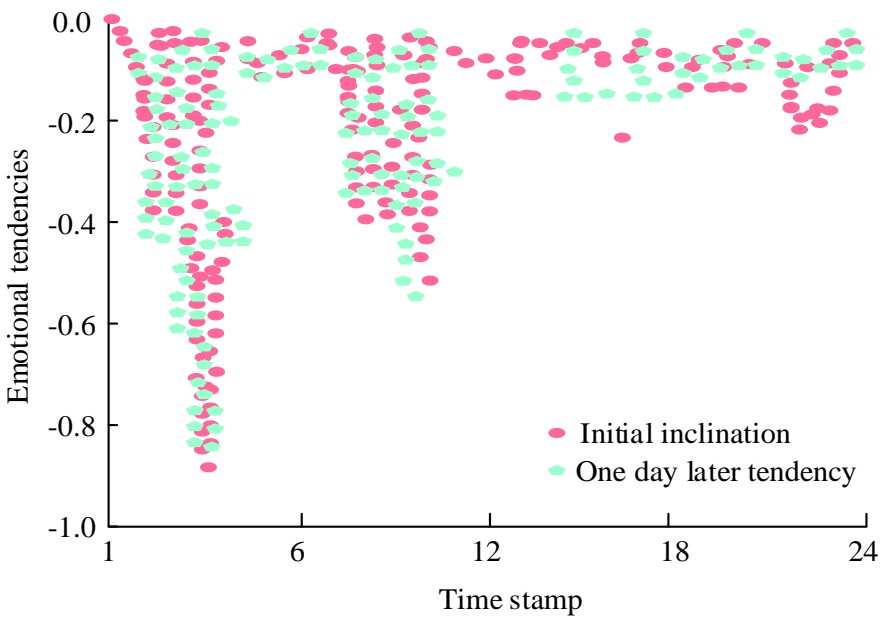

**Figure 10** Emotional tendency map of the topic "Israel demands millions of people from the north of Gaza to move to the south".

pertaining to the topic "Israel demands millions of people from northern Gaza to migrate to the south" is illustrated in Fig. 10.

Figure 10 presents the emotional orientation map concerning "Israel's demand for millions of people from northern Gaza to migrate to the south." Initially, as the incident unfolded, the public expressed their perspectives with an expectation of Israel's handling being reasonable, hence marking a low point in emotional inclination around the fifth timestamp. However, with Israel's relocation request, public sentiment surged dramatically. Yet, by the 10th timestamp, due to various complex factors, emotions experienced a rebound. The evolution of the comprehensive score reflecting the intensity of emotional responses to the topic is depicted in Fig. 11.

In Fig. 11, initially, there is a noticeable level of heat, but no comments are recorded, resulting in a score of 0. As time progresses, attention towards the events escalates, leading to a surge in comments and an overall increase in the score reflecting emotional intensity. By the third timestamp, when Israel makes the relocation request, the score reaches its peak value, indicating heightened emotional engagement. However, as the event unfolds further, the score gradually diminishes, suggesting a gradual decrease in emotional intensity over time.

## CONCLUSION

Emotional analysis and public opinion monitoring in the contemporary media landscape pose formidable challenges. The aim is to precisely discern public sentiments and trends in public opinion, thereby furnishing a foundation for informed decision-making. To

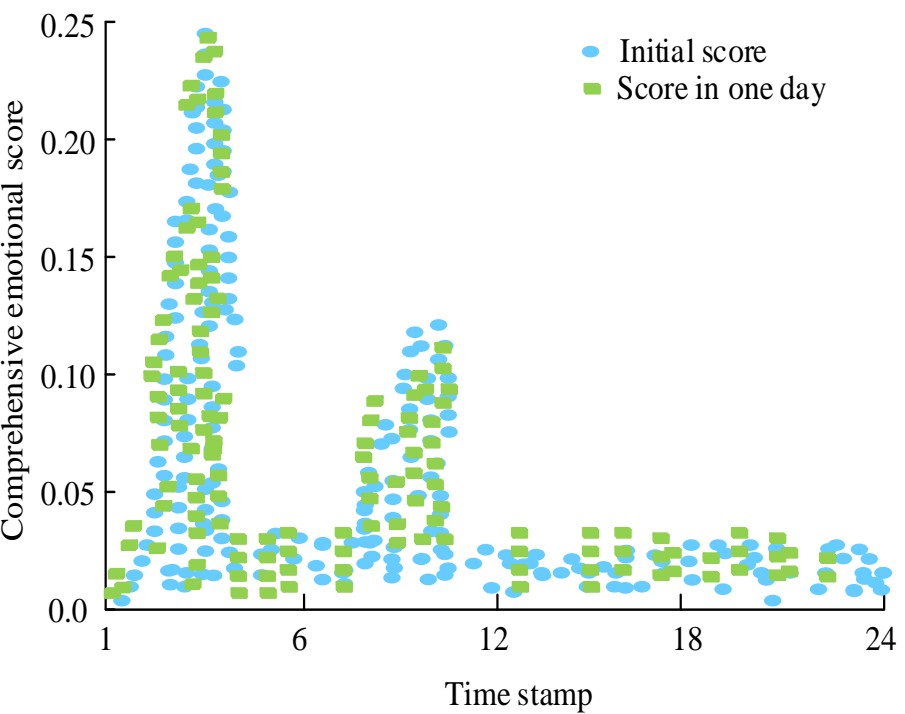

**Figure 11  Evolution chart of comprehensive score of topic emotional intensity.**

address this, a novel public opinion monitoring model leveraging deep learning technology has been devised. Results from three classification experiments closely mirror those of the two-classification scenario. The pinnacle of accuracy in emotional analysis of textual content stands at 90.8%, while the highest accuracy achieved with the parent dataset reaches 89.0%. Notably, emotional recognition in video content initially registers the lowest accuracy. However, through model optimization efforts, this accuracy is notably enhanced, reaching 6.74% and 6.97%, respectively.

Meanwhile, deep learning models offer the capability to track and analyze the evolving emotional intensity of events in real-time, thereby facilitating the monitoring of shifts in public opinion dynamics. Using Israel's migration request as an illustrative example, the model adeptly captures the progressive increase in the comprehensive score reflecting emotional intensity over time, peaking at the third timestamp. The significance of this research lies in the introduction of a novel media sentiment analysis and public opinion monitoring model grounded in deep learning principles. This model enables real-time monitoring of public opinion shifts while upholding a high level of accuracy, thereby holding substantial implications for social governance and decision-making processes. Nevertheless, owing to constraints inherent in the training dataset, the model may encounter occasional misjudgments when confronted with emotionally nuanced or complex scenarios. Consequently, future research endeavors will focus on refining

the model, enhancing its generalizability and accuracy, thereby enabling its broader applicability across diverse scenarios.

### Funding

This article is funded by the school-level project of Guangzhou Huashang College, "Study on the stigmatization based on big-data of social media in the post-truth era" (Project No.: 2020HSDS12). The funders had no role in study design, data collection and analysis, decision to publish, or preparation of the manuscript.

### Grant Disclosures

The following grant information was disclosed by the author:
The school-level project of Guangzhou Huashang College, "Study on the stigmatization based on big-data of social media in the post-truth era": Project No.: 2020HSDS12.

### Competing Interests

The authors declare there are no competing interests.

### Author Contributions

- Haili Xu conceived and designed the experiments, performed the experiments, analyzed the data, performed the computation work, prepared figures and/or tables, authored or reviewed drafts of the article, and approved the final draft.

### Data Availability

   The raw data and code are available in the Supplementary Files.

### Supplemental Information

Supplemental information for this article can be found online at http://dx.doi.org/10.7717/peerj-cs.2069#supplemental-information.

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
