# Peer review of "A bigura-based real time sentiment analysis of new media"

_PeerJ Computer Science, doi:10.7717/peerj-cs.2069_

## Round 0.1 · original submission · Major Revisions

Dear Author Thank you for your submission. We have carefully reviewed your article and the domain experts have also evaluated the quality of the manuscript. Generally, they are satisfied with the contributions and quality of the the article. but also suggesting a number of improvement areas. Please consider their comments and mine below. You are requested to carefully revise the paper in light of these comments and resubmit for further consideration.

AE Comments: Please justify/outline the novelty of work clearly in introduction section.
The technical language of the paper needs improvement.
References must be consistent in a single pattern.
As there is one author and affiliation, therefore, numbering is not needed in the header section.
Mostly Equation numbers are missing.
Please define all the acronyms first time and then use onward
Please re-adjust the caption of Table 1 for more clarity.

**Language Note:** The Academic Editor has identified that the English language must be improved. PeerJ can provide language editing services - please contact us at [email protected] for pricing (be sure to provide your manuscript number and title). Alternatively, you should make your own arrangements to improve the language quality and provide details in your response letter. – PeerJ Staff

·

Basic reporting

Regarding sentiment analysis for new media, this study emphasizes the clear benefits of deep learning over more conventional techniques like Bayesian and K-Nearest Neighbor classifiers. Deep learning shows itself to be a powerful tool for deciphering users' emotional inclinations in the digital sphere, with increases in classification accuracy of 3.88 percent and 4.33 percent, respectively.
However, the following suggestions can further improve the quality of the article.
The utilization of deep learning in this study for sentiment analysis and public opinion monitoring in the new media landscape has yielded promising results. With accuracy rates reaching as high as 92.7% for text content emotions and 86.9% for video content emotions, it's evident that deep learning models excel in capturing the nuances of user emotions. However, the claim made by the authors from lines 116-119 about attentional convolutional and conditional fields is not entirely correct in emotion detection from real-time data since you need predefined conditional fields.
By providing new technical means for sentiment analysis and public opinion monitoring in new media, this research contributes significantly to the field. Its potential for achieving more accurate and real-time monitoring of public opinion carries important implications for ensuring social stability and public safety in an increasingly interconnected world. It is recommended to address the following

1. The introduction appears to be lacking in depth due to limited number of citations, which may compromise the credibility of the Related Work. Incorporating more citations from relevant scholarly sources would strengthen the foundation of the research.
2. Writing the introduction for a single paragraph may not present the significance of the study. Moreover it lacks in highlight gaps in existing research. A more detailed introduction will better engage readers and present better understanding for the subsequent sections of the paper.
3. None of the citations in the introduction section are integrated into the Related Work. It is important to ensure consistency between the introduction and the subsequent Related Work to maintain the flow of the paper.
4. Abbreviations are used without preceding phrases. Consider writing full phrase upon first use to enhance clarity and accessibility throughout the manuscript. This ensures that readers unfamiliar with the terminology can easily understand the text. for example GWO, WOA at line number 62, RECOLA at line 72.
5. In related work, there is an issue with paragraph organization, characterized by lengthy and complex passages. Implementing clearer separation and refining paragraph structure would greatly enhance the readability of the Related Work, facilitating improved understanding for readers.
6. In research method section, the citation 16, 17 on line 121 are not discussed in the related work, it could mean that important information might be missing from the main discussion. It's important to include all relevant citations in the related work so that readers get a complete picture of the research background.

Experimental design

1. "ACNNC" is not a widely recognized acronym in the context of new media. Could you please provide more details about 'ACNNC' assuming this is a specific term or concept, mentioned at line 122? It’s mentioned that the ACNNC model is composed of five levels. However, no description of these levels is provided. It would be beneficial for the reader to have more description or example on the five levels of the model you mentioned. Providing an example for each level and layer of the model would enhance understanding and clarify the models application within the context of your study.
2. The dataset of weibo is used but the structure of the dataset is not mentioned, at one point the author is explicitly defining the term word-vectors then manuscript is using videos and images too. A preprocessing of dimensions is entirely missed.

Validity of the findings

1. Figure 5 BiGura Model {softmax and concat layers are not attention layers}
2. The same claim has not citation too.
3. Since the conditional probability is being used then the emotion labels must be indicated first.
4. CRF layers inputs are not consistent with the system requirements, try considering normalization of 128 dimensions to 16 dimension. It may improve computational time, space and impact on result will be optimal.
5. A serious concern about the authenticity of this work is utility of Naïve Bayse for training for attention layer, this conditional probability is directed towards binary emotion (Emotions are not itself binary in essence) May I suggest using Self-attention training instead of conditional training process.
6. The deployment of GRU layers and its clever setting needs proper demonstration, why GRU layers are performing in binary styles.

Additional comments

1. The manuscript refers to references 19 and 20, yet these sources are not cited within the main text. Consider adding a few additional references to strengthen the literature base and provide further support for the arguments presented in the manuscript.

Cite this review as

·

Basic reporting

.

Experimental design

.

Validity of the findings

.

Additional comments

This study provides new technical means for sentiment analysis and public opinion monitoring in new media. It helps to achieve more accurate and real-time monitoring of public opinion, which has important practical significance for social stability and public safety. This study definitely contributes to the body of knowledge. However, there are some major and minor changes are required to enhance its legibility.

1. The title is not expressive enough, it does not tell information about the methodology employed i.e. Innovative deep learning may be replaced with the term BiGRUA framework.
2. This manuscript is all about real-time public opinion monitoring, may be mentioned in the title of the paper
3. The abstract of this manuscript needs rewriting as there are multiple terms either misleading or not covering the subject matter such as “New Media”/news media, or “Deep learning technology” etc. It also failed to provide enough information about the research problem, methodology employed etc.
4. There are multiple grammatical errors in the manuscript
5. Paragraph from line 43-49 needs attention many key points mentioned but none of the attended properly in the manuscript such as controlling public opinion risk, research perspective methodological bibliography. Even term shortcomings is mentioned, however, there is no section or subsection that tells about shortcomings of this research idea
6. Line 54 citation missing
7. Line 59 citation missing
8. SENResNet terminologies used more than once but never defined in the lines 81-83 etc.
9. Related work needs additional references that explicitly dealing with real time emotion detection
10. Figure 1, source is missing cite the contributor
11. Figure 3 word segmentations is vivid many details are missing, consider redrawing

Cite this review as

---

## Round 0.2 · accepted · Accept

Dear Author,
Thank your submission to our esteemed journal, based on the input from the experts, The revised version of the paper is quite improved and acceptable for publication as suggested by the experts. Therefore, we are pleased to notify you the acceptance of the article. However, I suggest the following during your final proofs/files submission .

Please make the title a little more elaborative and include the words of deep learning etc. the techniques you have used in the paper.
Thank you

·

Basic reporting

The basic reporting aspects of the manuscript are in accordance with the requirements.

Experimental design

The the experimental design is satisfactory.

Validity of the findings

.

Additional comments

The paper has been updated in light of the previous comments.

Cite this review as

·

Basic reporting

comment are incorporated and authors have updated the papers. Therefore, paper is accepted.

Experimental design

comment are incorporated and authors have updated the papers. Therefore, paper is accepted.

Validity of the findings

comment are incorporated and authors have updated the papers. Therefore, paper is accepted.

Additional comments

comment are incorporated and authors have updated the papers. Therefore, paper is accepted.

Cite this review as